# Olive Leaf Tea Impact on Postprandial Glycemia: A Randomized Cross-Over Trial

**DOI:** 10.3390/foods12030528

**Published:** 2023-01-24

**Authors:** Manuela Meireles, Anna Carolina Cortez-Ribeiro, Denise Polck, Juliana Almeida-de-Souza, Vera Ferro-Lebres

**Affiliations:** 1Centro de Investigação da Montanha (CIMO), Instituto Politécnico de Bragança, (Mountain Research Center—Polytechnic Institute of Bragança), Campus Santa Apolónia, 5300-253 Bragança, Portugal; 2Laboratório para Sustentabilidade e Tecnologia em Regiões de Montanha, Instituto Politécnico de Bragança, (LA SusTEC—Associated Laboratory for Sustainability and Technology in Inland Regions, Polytechnic Institute of Bragança), Campus Santa Apolónia, 5300-253 Bragança, Portugal; 3Polytechnic Institute of Bragança, Campus Santa Apolónia, 5300-253 Bragança, Portugal

**Keywords:** olive leaves, blood glucose, postprandial glycemia, diabetes

## Abstract

This study investigates the effect of olive leaf tea (OLT) on postprandial glycemia in healthy volunteers when ingested with a high-carbohydrate meal, compared with a placebo tea (CON). Healthy adults participated in a double-blind, randomized, placebo-controlled, and cross-over design trial receiving a high-rich carbohydrate meal with either 250 mL of OLT or CON at two different times after a washout period. The sequence order was randomized at a ratio of 1:1. Capillary blood glucose was measured in a 2 h period after ingestion. Eighteen participants were initially randomized. Of these, thirteen completed the trial and were analyzed. The consumption of OLT resulted in a delay in peak time (48.5 ± 4.2 min vs. 35.7 ± 4.0 min, *p* = 0.03) and a significant increase in glucose area under the curve compared to placebo (14,502.7 ± 640.8 vs. 13,633.3 ± 869.4 mg/dL·min, *p* = 0.03). Results are depicted as mean ± SEM. The OLT and CON palatability were generally well accepted. No adverse effects were reported. OLT did not ameliorate a glycemic curve induced by carbohydrate-rich meal ingestion, suggesting that at least when ingested acutely in a single meal, OLT does not have antihyperglycemic effects. Future studies should account for chronic consumption providing a better understanding of glycemic regulation over time.

## 1. Introduction

Diabetes is the ninth leading cause of death in the world [1]. Approximately 537 million people live with it daily [1,2], and it represents an increased risk of heart attacks and strokes [3,4]. Diet is one of the most effective interventions to prevent and treat glycemic dysregulation among individuals with type 2 diabetes [5], yet one of the most difficult to manage [6], resulting in many people worldwide being at increased risk of having diabetes or already having the criteria for a prediabetes diagnosis.

The quest for food products with antihyperglycemic properties represents a significant proportion of diabetes research. Olive leaf infusions have been used in traditional herbal medicine as a way to treat and prevent many diseases, including diabetes [7]. In addition to its potential anti-glycemic effects, finding therapeutic value in olive tree leaf also has a sustainable dimension. Olive tree (*Olea europaea*, Oleaceae) leaf waste is generated during the mechanical harvesting of the olive fruit and after the necessary pruning to maintain the tree’s health [8]. This waste resulting from olive oil production is rich in secoiridoids, a class of polyphenols mainly found in olive oil and the olive tree leaf [9]. Oleuropein, the main secoiridoid found in this leaf is known to have many biological activities, including action on incretins, insulin secretion, and glucose uptake, thus helping to regulate glycemia [10]. However, this anti-glycemic effect still lacks robust scientific evidence in humans [10].

High-glycemic diets can potentially increase the risk of developing diabetes [11]. Pairing high-glycemic meals with beverages that may decrease their glycemic impact has been studied as an option to minimize health effects [12,13]. Therefore, this study investigates the effect of olive leaf tea on postprandial glycemia in healthy volunteers when ingested with a high-carbohydrate meal, compared with a placebo tea. This study hypothesized that olive leaf tea would improve glycemic control and modulate postprandial glycemia. Evidence resultant from this study reveals that at least in young healthy adults, and in acute conditions, this effect was not verified.

## 2. Subjects, Materials, and Methods

### 2.1. Participants

Participants were recruited among dietetics and nutrition trainers at the Polytechnic Institute of Bragança or their colleagues. To be considered eligible, participants had to be between 18 to 60 years old, not pregnant, and not taking any drugs that could interfere with glucose metabolism. The purpose of this project was explained to each voluntary participant before they signed an informed consent. To maximize data protection and to comply with the General Data Protection Regulation all the information was collected anonymously. On the day of data collection, a unique code number was assigned to each participant which was saved to allow for cross-checking with the allocated treatment on the second visit, guaranteeing the anonymity of the process. This study was conducted in accordance with The Code of Ethics of the World Medical Association (Declaration of Helsinki) for experiments involving humans and had ethical approval from the Ethic Commission of Institute Polytechnic of Bragança (nº80/2022).

### 2.2. Experimental Design

The study had a double-blind, randomized, placebo-controlled, and cross-over design (Appendix A) and followed the Consolidate Standards of Reporting Trials (CONSORT) 2010 statement [14].

During the first visit, participants were randomly allocated to either olive leaf tea (OLT) or placebo (CON) at a ratio of 1:1 using a randomization scheme generated by an online randomization tool for researchers, available at the Website Randomization.com (https://www.randomization.com/ accessed on 28 April 2022) (seed 12097). Each participant completed a wash-out period of at least 7 days before the second round, where those who had been assigned tea during the first visit received a placebo, and those who were assigned a placebo at the first round received tea.

This research took place at the nutrition laboratory of the Health School at the Polytechnic Institute of Bragança, Portugal. Recruitment, data collection, and data analysis were performed between April and July 2022. Sample size was determined based on previous publications of cross-over trials measuring postprandial glucose responses [15,16] and sample size power confirmed by post-hoc analysis.

### 2.3. Pre-Trial Procedures

Participants were asked to arrive at the laboratory at 8.45 h in the morning after an overnight fast of 12 h. Water was allowed during the fasting but not during the trial. In addition, participants were asked to avoid olives or olive oil consumption in the last meal of the previous day and abstain from physical exercise in the morning before the trial. Anthropometric data was gathered and characterization of the participants (age, sex, body mass index) was performed right before the beginning of the trial.

### 2.4. Test Products

Dried olive leaves were obtained from a local company of biological products “+ervas” (www.maiservas.com–accessed on 21 April 2022). Infusions were prepared according to the manufacturer’s instructions, by adding 250 mL of water at 90 °C to the product, agitating, and waiting for 5 min. The researcher who prepared the teas was not involved in the outcome assessments.

The amount of product was chosen considering its palatability. The dose of 1 g of chopped leaves per tea was selected by the research team after an initial pilot sensory trial. This pilot was conducted with 7 people testing the infusion with either 1 g or 2 g, where 5 out of 7 preferred the first option. The placebo tea was prepared using a food coloring, E150a, in the same way as previous authors [17]. The test products were offered as part of a standardized, carbohydrate-rich, mixed breakfast meal containing 110 g of wheat bread and 50 g of peach jam. The composition of the mixed breakfast meal is characterized in Table 1.

### 2.5. Glycemic Measurements

The pre-established meals containing either olive leaf tea or the placebo were provided after the initial glycemic measure (t = 0) and consumed in no longer than 15 min. Capillary blood glucose was measured at times 0, 15, 30, 60, 90, and 120 min using the GlucoMen(R) aero2k system. Both participants and researchers collecting data were blind to the allocation treatment.

### 2.6. Data Analysis

Area under the curve (AUC) was calculated using the trapezoid rule. Normality was tested using Shapiro–Wilk test. Two way-ANOVA was performed to detect if the order of allocation or sex was a source of variation. Differences between treatments on the variables including glycemic peak, time to peak, and AUC were tested using the Wilcoxon matched-pairs signed rank test. Results were analyzed using GraphPad Prism 8.02. A post-hoc power calculation was performed using an online tool available at https://clincalc.com/stats/Power.aspx. Considering AUC as the primary endpoint and a 95% confidence interval, this study’s sample size had a power of 82.7%.

## 3. Results

### 3.1. Participant Characteristics

Of 52 participants invited to participate in the study, a total of 25 participants manifested interest. Eighteen participants were eligible and randomized to enroll in the study between May and July 2022. Of those, two failed to be present at the first round, one due to illness, and another could not attend due to a schedule change. After two extra drop-outs in the second round (coincident with the end of the study term), 14 participants completed the two rounds of the study (Appendix A).

One participant had been diagnosed with diabetes. Since this was an isolated case, we could not compare their glycemic response with healthy individuals without a diabetes diagnosis, and we decided to exclude this participant. Diabetes was not initially an exclusion criterion, leaving the opportunity to compare subgroups of the population, which was not possible due to this being an isolated case. Curiously and despite the AUCs in both rounds of this participant being much higher than the rest of the participants, we found that his response to the tea when matched to his placebo’s response was coincident with the global response of the participants. Notwithstanding this, the exclusion of this case decreased bias when looking at peak (max) or time to peak. We also excluded data from the participants who could not be present in the second round.

The thirteen participants analyzed were aged between 18 and 38 years old and had a body mass index (BMI) between 18.97 kg/m^2^ and 41.16 kg/m^2^ (Table 2). Except for one participant being obese, all the participants had a normal weight (BMI between 18.5 kg/m^2^ and 25 kg/m^2^).

The fasting plasma glucose values of all participants were normal—<100 mg/dL—with the exception of one participant who had a fasting glucose of 107 mg/dl in the first round. We decided not to exclude this participant since his metabolic response did not differ significantly from other participants. Additionally, this participant had normal glycemia in the second round.

### 3.2. Tolerability of Tested Products

The OLT and control tea palatability were generally well accepted. No adverse effects were reported for any of the meals tested.

### 3.3. Effect of Olive Leaf Tea on Postprandial Glycemic Responses

At baseline, there were no significant differences between the capillary plasma glucose measurements before the CON or OLT interventions (80.5 ± 2.4 mg/dL vs. 83.5 ± 3.8 mg/dL). We tested if the allocation of the order of treatment (CON-OLT vs. OLT-CON) interacted with the effects of the treatment on glucose AUCs (*p* > 0.05). Since an interaction effect was not observed, the following analyses were performed disregarding allocation order. Despite not being the primary focus of this study, a sex interaction was also tested and there were no significant differences between males and females (*p* = 0.75).

The consumption of OLT resulted in a significant increase in glucose area under the curve compared to the placebo (Figure 1A,B), and a delay in the peak glucose time (48.5 ± 4.2 min vs. 35.7 ± 4.0 min) (Table 3). Despite the delay of peak time, the peaks were not different between interventions (142.5 ± 7.6 mg/dL vs. 146.5 ± 8.4 mg/dL). No significant differences (*p* < 0.05) between conditions at the individual time of glycemic response were noted (Figure 1C).

## 4. Discussion

This randomized cross-over clinical trial tested the postprandial glycemic response to a high carbohydrate-rich meal when ingested with olive leaf tea (OLT) compared with a control beverage. Participants in our study were healthy and had a predictable glycemic response. We observed that in both situations there was a rapid increase in glycemic response, with glycemic peaks being reached between 30 and 60 min after the beginning of the trial. To our knowledge, this is the first study that analyzes the effect of OLT on glycemic response in an acute RCT.

Area under the curve is routinely used in the oral glucose tolerance test (OGTT). By measuring the rise of glycemia during OGTT, it is possible to detect impaired glucose tolerance [18]. Recently, mixed meal tolerance tests (MMTT) have been presented as alternatives to oral glucose tolerance tests (OGTT) with strong correlations between glucose values following both tests [19]. In this study, the mixed breakfast meal (bread with peach jam and OLT/CON tea) also had good acceptability for participants. The cross-over design of this study is one of its strengths, allowing us to individually compare the difference between AUC after the tested beverage against AUC after the placebo. According to our hypothesis, we would expect that OLT would result in lower AUCs, which was not the case. This hypothesis was based on a few randomized studies that tested the health effect of olive leaves in humans. In general, these studies used extracts from olive leaves, and only one study has previously studied olive leaf tea [20]. The first study with olive leaf extract (OLE) was made in 2012 by Wainstein and colleagues who gave 500 mg of OLE to patients with diabetes for 14 weeks and saw a decrease in HbA1c and improvements in glycemic response [21]. However, one year later, de Bock and colleagues first conducted a well-designed randomized crossover trial considering diet and physical exercise and adjusting the results in a way to guarantee the independence of OLE’s effect. In their study, the OLE intervention group had a 15% improvement in insulin sensitivity and a 28% improvement in pancreatic b-cell responsiveness compared to a placebo. This effect was seen after 12 weeks of supplementation with a well-characterized OLE in 46 middle-aged overweight men [22]. However, even when using extracts, the results were contradictory and no significant changes were observed for glucose or insulin levels after 4 weeks of 500 mg OLE supplementation in 77 healthy adult overweight or obese subjects, compared to a placebo [23]. Araki and colleagues were the only ones, to our knowledge, to test the effects of natural olive leaf tea instead of OLE. In their 12-week intervention with OLT, a decrease in fasting plasma glucose was shown, but with no changes in insulin levels or HbA1c [20]. None of these studies examined acute effects in a postprandial context and our results are not aligned with an increase in insulin sensitivity induced by OLT.

In the current study, OLT ingestion induced a significant delay in the postprandial glucose peak. This effect could be caused by slower digestion of starch, which would be consistent with an anti-α-amylase effect previously attributed to an olive leaf extract [24]. Nevertheless, a delayed glucose peak time has been associated with a decrease in insulin secretion and insulin sensitivity which is consistent with an increase in area under the curve during the 2 h period [25].

The phenolic content of olive leaves can be influenced by cultivar, agronomical factors, climatological conditions, and edaphology. Leaves from the same cultivar and geographical production were previously characterized as having, besides oleuropein: hydroxytorosol, chlorogenic acid, cafeic acid, verbascoside, rutin, apigenin-7-o-glucoside, and luteolin [26]. Variations in olive leaf tea chemical composition may exert variations in the observed biological activities [27]. The phenolic content present in olive leaf has similarities with the unsaponifiable fraction of olive oil. By using olive leaf tea, we can exclude the effects attributed to components of the saponifiable fraction of olive oil, such as the oleic acid [28]. Olive leaf tea is an alternative to ingest the phenolic content in a non-energetic product at meals where olive oil is not usually present.

One limitation of our study could be the duration of 120 min. Although 120 min is the reference for OGTT and commonly used in postprandial glycemic analyses, an extension of the measurements until 150 or 180 min would have added extra information and a more comprehensive analysis of the full effect of OLT beyond the 2 h interval. Our sample is relatively small to perform powerful subgroup analysis; however, one cannot exclude the possibility of a sex-based difference in glycemic regulation [29]. Another limitation to be taken into account is that results from this trial represent the effect on a small sample of a young healthy population living in a particular region and extrapolations from this trial to the general or non-healthy population should be taken carefully.

## 5. Conclusions

One of the major purposes of this study was the possibility of finding nutritional value in the use of olive leaf tea and at the same time finding dietary strategies to help prevent diabetes. The results from this study were not aligned with this main goal. Olive leaf tea did not ameliorate a glycemic curve induced by carbohydrate-rich meal ingestion; however, OLT delay of the glycemic peak should be further explored. Additionally, further studies should account for chronic consumption to provide a better understanding of glycemic regulation over time with an evaluation of the impact on HbA1c and other metabolic biomarkers.

## Figures and Tables

**Figure 1 foods-12-00528-f001:**
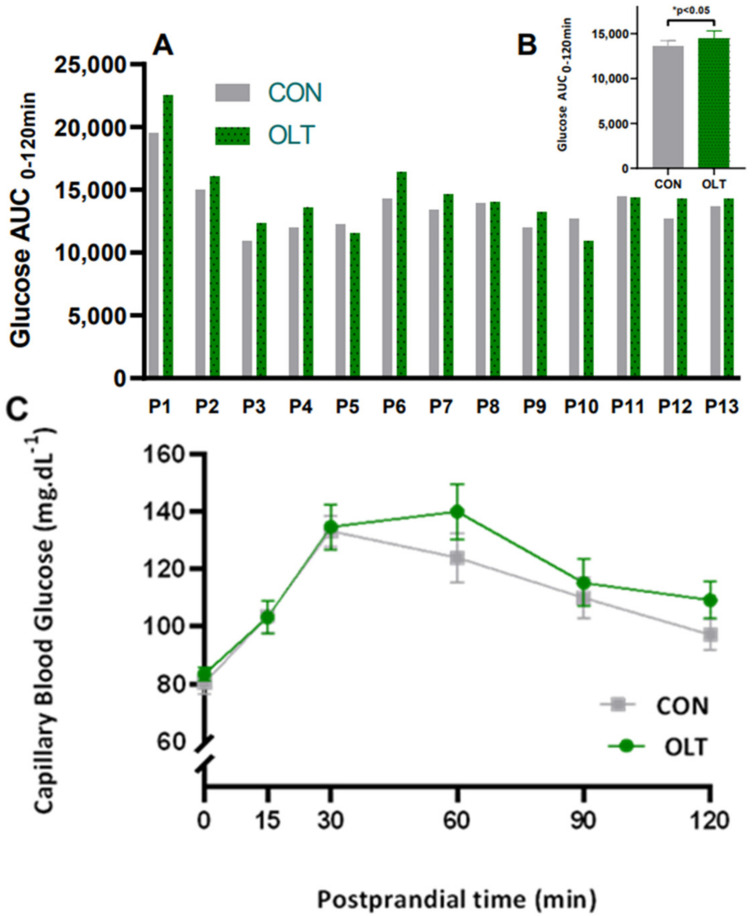
Individual responses (**A**) and summary mean (**B**) of postprandial glucose AUC from 0 to 120 min of 13 subjects and postprandial capillary plasma glucose from 0 to 120 min (**C**) after a mixed meal intervention composed of 2 slices of wheat bread (110 g) and 50 g of peach jam with olive leaf tea (OLT) or two slices of wheat bread (110 g) and 50 g of peach jam with 250 mL of placebo tea (CON). Glucose AUCs after OLT are significantly higher than CON. * *p* < 0.05 obtained after Wilcoxon matched-pairs signed rank test.

**Table 1 foods-12-00528-t001:** Composition of the mixed breakfast meals.

	CHO (g)	Protein (g)	Total Fats (g)	Energy (KCal)
MEAL 1—OLT				
White Bread 110 g	49.5	10.8	3.0	277.2
Peach Jam 50 g	24.0	0.3	3.0	98.0
Olive leaf tea 250 mL	0.0	0.0	0.0	0.0
Total	74.5	11.1	3.0	375.2

MEAL 2—CON				
White Bread 110 g	49.5	10.8	3.0	277.2
Peach Jam 50 g	24	0.3	0.0	98
Placebo tea 250 mL	0.0	0.0	0.0	0.0
Total	74.5	11.1	3.0	375.2

OLT: Mixed meal intervention containing olive leaf tea. CON: Mixed.

**Table 2 foods-12-00528-t002:** Baseline characterization of participants.

	Total Participants
Age Mean (SD)	24.2 ± 6
Sex	
Female—*n* (%)	9 (69.2%)
Male—*n* (%)	4 (30.8%)
BMI Mean (SD)	23.4 ± 5.7
FPG Mean (SD)	82.0 ± 3.2

FPG—Fasting Plasma Glucose; BMI- Body Mass Index.

**Table 3 foods-12-00528-t003:** Glycemic responses of the two separate interventions.

	CON	OLT	*p*-Value
Mean baseline glucose (mg/dL) ± SEM	80.5 ± 2.4	83.5 ± 3.8	0.52

Glycemic peak (mg/dL) mean ± SEM	142.5 ± 7.6	146.5 ± 8.4	0.93
Change (%)	REF	6.3 ± 4.1	

Time to peak (min) mean ± SEM	35.7 ± 4.0	48.5 ± 4.2	0.03
Time to peak (min) median (IC)	30 (30–45)	60 (30–60)	*p* < 0.001
Change (%) †	REF	50 ± 16.0	

Glycemic AUC (mg/dL·min) mean ± SEM	13,633.3 ± 869.4	14,502.7 ± 640.8	0.03
Change (%) †	REF	6 ± 2.4	

† Baseline fasting glucose, peak blood glucose concentration, time to peak, and incremental postprandial areas under the curve (AUC) from 0 to 120 min after test infusion meals composed of 2 slices of wheat bread and 50 g of peach jam with olive leaf tea (OLT) or two slices of wheat bread and 50 g of peach jam with 250 mL of placebo tea (CON). † Calculated as mean of the individual % of the changes from the control meal (REF).

## Data Availability

The data presented in this study are available on request from the corresponding author.

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
