# Peer review of "Olive Leaf Tea Impact on Postprandial Glycemia: A Randomized Cross-Over Trial"

_foods, 2023, doi:10.3390/foods12030528_

Round 1

Reviewer 1 Report

1-    The idea of the study is not new and its results are not sufficient to rely on it

2-    Too many biological factors affecting the results of this study make it unhelpful

3-    Controls should have been done to compare the results, such as using natural extracts or compounds that have a clear and known effect on blood sugar

4-    The study should have been conducted on a specific age group to produce consistent and confirmed results

5-    The study had to be done on a large number of volunteers, in addition to using a rather long period of time

Reviewer 2 Report

The current study is intriguing, although the neutral effects and the fact that some data are missing. In addition, the authors need to report something about the chemical analysis of the composition of olive-leaf tea.

The health benefits attributed to the consumption of olive oil are ascribed to its balanced composition, which can be divided into two main fractions: a saponifiable fraction (98% of the total content) and an unsaponifiable fraction (2%). The last minor fraction is formed by phenols, tocopherols, phytosterols, volatile compounds, terpenes, and hydrocarbons. This little fraction is representative of olive oil, and most of these families are lost during the refining process.

Moreover, in addition to the cultivar, the phenolic content depends on agronomical factors, climatological conditions, edaphology, and technological methods used for oil extraction.

So, do the authors know the chemical composition of the olive leaf tea extract?

 The authors should add a paragraph with a discussion about the content of olive leaf tea in the current study (maybe hypothetically if they did not know) in comparison with the composition of olive leaf tea from other studies.

Is there a better beneficial effect on glycemia from olive leaf tea over the use of olive oil?

The methodology is appropriate. The manuscript is well written, and the discussion/conclusions are acceptable. 

Overall, data could be of interest and future research if the authors add some comments on the composition of olive leaf tea.

Reviewer 3 Report

The manuscript “Olive leaf tea impact on postprandial glycemia: A randomized

cross-over trial” is a double-blinded, randomized, placebo-controlled, and cross-over aimed to investigates the effect of olive leaves tea (OLT) on postprandial glycemia in 18 healthy volunteers when ingested with a high-carbohydrate meal compared with a placebo tea (CON). The subjects received 250 ml OLT or CON at two different times after a washout period.

OLT did not ameliorate a glycaemic curve induced by carbohydrate-rich meal ingestion, suggesting that at least when ingested acutely in a single meal OLT does not have antidiabetic effects.

There are major issues to be clarified:

Comments to the Authors:

1.                   The overall language used in the manuscript needs improvement

2.                   Line 21, is there units for the results

3.                   Line 25, please use the expression antyhyperglicemic, not antydiabetic effect.

4.                   Line 34, …impacting mortality after acute myocardial infection…please delete this part of the sentence, it is not within the scope of this manuscript

5.                   Line 38 … the expression anti-glycemic does not exists, did you men antyhiperglycemic

6.                   Please, add more explanation in Methodology section about inclusion criteria, did you do 2h ogtt to exclude disturbance of glucose metabolism

7.                   In methodology section: please clarify the level of care (secondary, tertiary) where the subjects where the study was performed

8.                   Line 117, 118 please explain in details when the subjects have done SMBG in relation to drinks, before or after drinks consummation

9.                   Please add more limitation of the study, limited number of patients

10.               Line 152-153 prediabetes should not be diagnosed according to the values of SMBG

Reviewer 4 Report

This pilot study offers important insights into a potential dietary intervention and glycemic control. Often negative findings are not published; however, they are as valuable as positive findings. The study protocol was rigorous. Some of the English language is awkward; a copyedit by a native English speaker is recommended.

Line 15: Change “leaves” to “leaf”

Line 17: Change double-blinded to double-blind

Line 32: Add source to the opening sentence, if it is source 1 changed to …is the ninth leading cause…

Line 34:  Add one before of, diet is one of the…

Line 35: Add “among individuals with type 2 diabetes” …treat glycemic dysregulation among individuals with type 2 diabetes

Line 42: Change “leaves” to “leaf”

Line 53: Change “leaves” to “leaf”

Data analysis

Add power calculation

·        Sim J, Lewis M. The size of a pilot study for a clinical trial should be calculated in relation to considerations of precision and efficiency. J Clin Epidemiol. 2012 Mar;65(3):301-8.

Limitations

Add small sample size and potential note re low power size (<80%)

Add consideration of gender-based differences in glycemic control. For example, the findings of this study:

·        Choe SA, Kim JY, Ro YS, Cho SI. Women are less likely than men to achieve optimal glycemic control after 1 year of treatment: A multi-level analysis of a Korean primary care cohort. PLoS One. 2018 May 2;13(5):e0196719.

Round 2

Reviewer 1 Report

No comments

Reviewer 2 Report

no other comments

Reviewer 3 Report

The authors respond adequately, I do not have further questions.